# Tpr Misregulation in Hippocampal Neural Stem Cells in Mouse Models of Alzheimer’s Disease

**DOI:** 10.3390/cells12232757

**Published:** 2023-12-01

**Authors:** Subash C. Malik, Jia-Di Lin, Stephanie Ziegler-Waldkirch, Stefan Tholen, Sachin S. Deshpande, Marius Schwabenland, Oliver Schilling, Andreas Vlachos, Melanie Meyer-Luehmann, Christian Schachtrup

**Affiliations:** 1Institute of Anatomy and Cell Biology, University of Freiburg, 79104 Freiburg, Germany; subash.chandra.malik@ki.se (S.C.M.); jia-di.lin@anat.uni-freiburg.de (J.-D.L.); deshpande03@gmail.com (S.S.D.); 2Faculty of Biology, University of Freiburg, 79104 Freiburg, Germany; 3Department of Neurology, Medical Center, Faculty of Medicine, University of Freiburg, 79106 Freiburg, Germany; stephanie.waldkirch@uniklinik-freiburg.de (S.Z.-W.); melanie.meyer-luehmann@uniklinik-freiburg.de (M.M.-L.); 4Institute of Surgical Pathology, Medical Center, University of Freiburg, 79106 Freiburg, Germany; stefan.tholen@uniklinik-freiburg.de (S.T.); oliver.schilling@uniklinik-freiburg.de (O.S.); 5Institute of Neuropathology, University of Freiburg, 79106 Freiburg, Germany; 6Department of Neuroanatomy, Institute of Anatomy and Cell Biology, Faculty of Medicine, University of Freiburg, 79104 Freiburg, Germany; andreas.vlachos@anat.uni-freiburg.de; 7Center BrainLinks-BrainTools, University of Freiburg, 79110 Freiburg, Germany; 8Center for Basics in Neuromodulation (NeuroModul Basics), Faculty of Medicine, University of Freiburg, 79106 Freiburg, Germany

**Keywords:** adult neural stem cells, Alzheimer’s disease, nuclear pore complex, super resolution microscopy, translocated promoter region

## Abstract

Nuclear pore complexes (NPCs) are highly dynamic macromolecular protein structures that facilitate molecular exchange across the nuclear envelope. Aberrant NPC functioning has been implicated in neurodegeneration. The translocated promoter region (Tpr) is a critical scaffolding nucleoporin (Nup) of the nuclear basket, facing the interior of the NPC. However, the role of Tpr in adult neural stem/precursor cells (NSPCs) in Alzheimer’s disease (AD) is unknown. Using super-resolution (SR) and electron microscopy, we defined the different subcellular localizations of Tpr and phospho-Tpr (P-Tpr) in NSPCs in vitro and in vivo. Elevated Tpr expression and reduced P-Tpr nuclear localization accompany NSPC differentiation along the neurogenic lineage. In 5xFAD mice, an animal model of AD, increased Tpr expression in DCX+ hippocampal neuroblasts precedes increased neurogenesis at an early stage, before the onset of amyloid-β plaque formation. Whereas nuclear basket Tpr interacts with chromatin modifiers and NSPC-related transcription factors, P-Tpr interacts and co-localizes with cyclin-dependent kinase 1 (Cdk1) at the nuclear chromatin of NSPCs. In hippocampal NSPCs in a mouse model of AD, aberrant Tpr expression was correlated with altered NPC morphology and counts, and Tpr was aberrantly expressed in postmortem human brain samples from patients with AD. Thus, we propose that altered levels and subcellular localization of Tpr in CNS disease affect Tpr functionality, which in turn regulates the architecture and number of NSPC NPCs, possibly leading to aberrant neurogenesis.

## 1. Introduction

In the adult mammalian brain, the subgranular zone (SGZ) of the dentate gyrus in the hippocampus and the subventricular zone (SVZ) lining the lateral ventricles possess neural stem/precursor cells (NSPCs) that give rise to new neurons throughout life [1,2,3,4]. Newly generated cells in the SGZ are functionally integrated into the granule cell layer (GCL) and contribute to hippocampus-dependent learning and memory and, more specifically, pattern separation [5,6,7]. Adult neurogenesis declines with age and is associated with cognitive decline in age-related neurodegenerative diseases, such as Alzheimer’s disease (AD) [8,9,10].

Aberrant nuclear pore complex (NPC) functioning is implicated in neurodegeneration [11,12,13,14,15,16]. NPC components can become mislocalized or altered in expression in neurodegeneration, such as in Huntington’s disease (HD), amyotrophic lateral sclerosis (ALS), and Alzheimer’s disease (AD) [17,18,19,20,21]. In AD, the neuropathological protein tau directly interacts with individual NPC components, leading to the disruption of NPC function and neurotoxicity [18]. However, whether the altered extracellular environment in AD also affects individual NPC components in NSPCs—potentially contributing to the decline in neurogenesis—is still poorly understood.

The NPC functions as the sole gatekeeper to the nucleus by regulating the bidirectional trafficking of macromolecules, such as proteins and RNA, between the nucleus and the cytoplasm during key cellular processes such as cell signal transduction and cell growth [22,23]. The NPC comprises component proteins called nucleoporins (Nups). Approximately two thirds of these Nups are structural or scaffold Nups that form the physical structure of the pore. The remaining one third are phenylalanine-glycine repeat Nups (FG-Nups) that form the selective permeable barrier within the nuclear pore [24,25,26,27].

The nuclear basket of the NPC includes three FG-Nups: Nup50, Nup153, and translocated promoter region (Tpr) [28]. Tpr has been described to have a role in HD, ALS and neurological disorder [11,17,29]; while its function in AD is not known yet. Tpr was initially described as having a role in nuclear protein export [30,31]. In addition, Tpr is a hub for NPC assembly, chromatin interaction and organization, mRNA export, and modulation of gene expression [32,33,34,35,36,37].

While the role of Tpr in NSPC differentiation is unknown, other Nups have been implicated in the regulation of neural stem cell fate. While Nup133 [38] and Nup 210 [39] regulate neural stem cell differentiation, Nup153 is known to maintain the neural stem cell pluripotency through gene silencing [40] and by acting as a transcriptional co-factor [41]. Thus, as described for other individual Nups [42], Tpr might also regulate NSPC cell identity and differentiation.

The contribution of individual Nups in the regulation of the altered neurogenesis in AD is only poorly described. Nup 153 interacts with Sox2 and regulates the transcriptional program in neural progenitor cells [41]. Altered Nup153 expression impairs the function of cultured hippocampal NSPCs isolated from AD mice and the restoration of Nup153 level was described to improve AD hippocampal NSPC behavior and their neuronal differentiation in vitro [43], suggesting Nups as novel targets for the restoration of physiological NSPC neurogenesis in AD. However, a role for Tpr in NSPC in AD has not been described.

We found increased Tpr expression in NSPCs along the neurogenic lineage in the two major stem cell niches of the adult mammalian brain, the SVZ and SGZ, but phospho-Tpr (P-Tpr) was localized to the nucleus only in doublecortin (DCX)+ neuroblasts. Aβ pathology in mouse models of AD and in human individuals with AD showed aberrant Tpr expression. In addition to interacting with macromolecular trafficking-related proteins, Tpr interacted with chromatin modifiers and transcription factors in NSPCs, and P-Tpr interacted with cyclin-dependent kinase 1 (Cdk1). Tpr expression regulated the NPC count in primary NSPCs, and misregulated Tpr expression in hippocampal NSPCs correlated with changes in the NPC count. Thus, we propose that altered Tpr expression levels and subcellular localization regulate the nuclear architecture in hippocampal NSPCs, which may contribute to the decline in neurogenesis in AD.

## 2. Materials and Methods

### 2.1. Animals

For examining Tpr and P-Tpr subcellular localization in NSPCs in vitro and in vivo, for interactome and immunoblotting studies, and for establishing Tpr expression level effects on NPC counts in NSPCs, adult C57BL/6 mice (8–12 weeks of age) were used in this study. For examining Tpr and P-Tpr in mouse models of AD, heterozygous 5xFAD transgenic mice at 6 weeks, 4 months, and 8 months of age, and APP23 transgenic mice at 9 months of age were used in this study. For 5xFAD and APP23 mice, age-matched non-transgenic littermates on a C57BL/6 background were used [44,45]. For the AD mouse model, only male mice were used because female mice have a faster and earlier onset of Aβ plaque formation. Therefore, to minimize variability and reduce sample size, only one gender was used. Experimental mouse numbers used are provided in the corresponding figure legends. Mice were kept under a 12 h light, 12 h dark cycle with food and water ad libitum.

### 2.2. Immunohistochemistry

Mice were transcardially perfused with ice-cold saline, followed by 4% PFA in phosphate buffer under ketamine and xylazine anesthesia, and brain samples were removed, cryoprotected, embedded in OCT (Tissue-Tek, Leica, Wetzlar, Germany), and frozen on dry ice. Brain samples were cut into 30-μm coronal sections with a cryotstat, and further processed for free-floating immunohistochemistry [46]. Briefly, sections were permeabilized with PBS-triton 0.3% for 30 min, blocked in 5% BSA for 1 h and incubated overnight with the primary antibody in PBS with 1% BSA. Primary antibodies used were rabbit anti-Tpr (1:1000, ab84516, Abcam, Amsterdam, The Netherlands), rabbit anti-phospho-Tpr (1:1000, S-2059, a kind gift of V.K. Nandicoori [47]), guinea pig anti-DCX (1:1000; Merck Millipore, Darmstadt, Germany AB2253), rat anti-Ki67 (1:300; Thermo Fisher Scientific, Darmstadt, Germany 14-5968-82), rabbit anti-APP (1:1000, Millipore, Darmstadt, Germany A8717), and mouse anti-NeuN (1:100; Merck Millipore, MAB377). Secondary antibodies were conjugated with Alexa Fluor 488 or 594 (1:200; Jackson ImmunoResearch Laboratories, Eching, Germany). Sections were cover-slipped with DAPI (Southern Biotechnology, Eching, Germany).

### 2.3. Super-Resolution Imaging

Zeiss LSM 880 Observer/Fast Airyscan with an inverted microscope was used for SR imaging of NSPCs in vitro and mouse 5xFAD tissue sections. A plan-apochromat 63×/1.4 oil objective was used with a zoom factor (10×) while imaging single cells. Airy units were calibrated before each session for the two different channels (green and red) where the fluorophores were targeted for acquisition. Additional Airyscan image processing was applied after confocal imaging to obtain the final Airyscan processed images.

### 2.4. NPC Quantification

Super resolution (SR) images obtained by Airyscanning were processed in IMARIS (Bitplane, Schlieren, Switzerland). In IMARIS, the ‘surface’ tool was used to draw, contour, and create the entire nuclear volume manually, keeping DAPI as the reference. Different z stacks were manually combined to obtain the whole nucleus. Next, the nucleus was masked with the commands ‘constant inside’ and ‘voxels outside to 0’. The spot tool used to define the dimension of the NPC was set at 150 nm and kept constant across all experiments. Spot detection was then automatically applied to obtain the total count of NPCs per nucleus. For each cell, the NPC count was normalized to the nuclear volume, as previously described [41,48].

### 2.5. Tpr Imaging in Human AD Tissue Sections

For human sections, paraffin-fixed human brains (male, 61–82 years) from healthy and AD (Braak IV-VI) individuals were used. For control 1 (61 years), the cause of death was cardiovascular failure and the comorbidities were kidney failure, diabetic nephropathy and prostate carcinoma. For control 2 (69 years), the cause of death was cardiovascular failure and the comorbidities were myelodysplastic syndrome and non-Hodgkin lymphoma. For control 3 (76 years), the cause of death was cardiovascular failure and the comorbidities were acute myeloid leukemia and gastrointestinal bleeding. For control 4 (82 years), the cause of death was cardiac shock and the comorbidity was chronic obstructive pulmonary disease. For control 5 (74 years), the cause of death was cardiac shock and the comorbidity was chronic cardiac failure. For the AD sample 1 (73 years, Braak IV), the cause of death was multiple organ failure and the comorbidities were kidney failure, urothelial carcinoma, and aortic aneurysm. For the AD sample 2 (73 years, Braak IV-V), the cause of death was cardiac shock and the comorbidities were myocardial ischemia and coronary sclerosis. For the AD sample 3 (76 years, Braak V), the cause of death was cardiac failure and the comorbidities were gastrointestinal bleeding and (microinvasive) colorectal cancer. For the AD sample 4 (75 years, Braak V), the cause of death was multi organ failure and the comorbidity was multiple myeloma. For the AD sample 5 (82 years, Braak VI), the cause of death was multi-organ failure and the comorbidities were septic shock, nephritis sclerosis, and stomach carcinoma. Brains from healthy and AD individuals were assessed according to the Braak AD staging procedure [49], and samples with no/unspecific Tpr immunohistochemical labeling were excluded. After sequential dehydration of 3-µm-thick tissue sections, an antigen retrieval step was performed in 0.1 M sodium citrate buffer (pH 6) at 110–120 °C with a pressure cooker. Sections were incubated with a primary antibody (rabbit anti-Tpr, 1:250, ab84516, Abcam) in 10% NDS and 5% BSA in 0.1 M glycine/PBST and incubated for 2 days.

A biotinylated antibody (1:200, 711-065-152, Jackson Immuno Research, Eching, Germany) was used to amplify the signal, followed by incubation with a streptavidin-AF594 antibody (1:1000, 016-580-084, Jackson Immuno Research). Sections were cover-slipped with DAPI (Southern Biotechnology). Imaging of single cells in the hippocampal SGZ was performed as described above with a Leica TCS SP8 confocal microscope. The quantification of overall Tpr expression around individual nuclei was performed as described [50]. Confocal images of these single cells were processed in IMARIS to obtain representative images of Tpr expression in single cells of the human hippocampal tissue sections.

### 2.6. Electron Microscopy

Mice were transcardially perfused with 0.05% glutaraldehyde in 4% PFA, 0.2% picric acid under ketamine and xylazine cocktail anesthesia, and brain samples were removed. Brain samples were cut into 50-µm sections on a vibratome. Sections were washed in 50 mM TBS, blocked with 20% normal goat serum, and incubated overnight at 4 °C with the primary antibody in PBS (50 mM TBS with 2% normal goat serum). Primary antibodies used were rabbit anti-Tpr (1:1000, ab84516, Abcam), rabbit anti-phospho-Tpr (1:1000, S-2059, kind gift of V.K. Nandicoori [47]). After washing in TBS for 1 h, sections were incubated with the second antibody overnight at 4 °C. The next day, silver intensification and DAB staining were conducted. Subsequently, the sections went through osmification, dehydration, and embedding in Durcupan, and ultrathin 60-nm sections were cut by a Leica UC 6. A Leo 906 E transmission electron microscope (Zeiss) with a sharp-eye 2 K CCD camera was used to acquire images at 2000× and 6000× magnification for the hippocampal SGZ and single cells in the SGZ. IPS software was used for image acquisition and Image J was used for quantification.

### 2.7. Adult Neural Stem/Precursor Cell Culture

SVZ-derived NSPCs were isolated and cultured as described previously [50,51,52]. SVZ-derived NSPCs were isolated to generate neurospheres [52]. Briefly, dissected adult mouse SVZ tissue was dissociated with 0.25% trypsin/HBSS, and cells were cultured at a density of 50,000 cells in 25-cm^2^ culture flasks in Neurobasal-A medium (Gibco, Darmstadt, Germany) containing B27 without vitamin A, Pen/Strep (1%), GlutaMax (1%), glutamine (0.5%), rhFGF2 (20 ng/mL) (all from Invitrogen, Darmstadt, Germany), and rhEGF (20 ng/mL; Sigma–Aldrich, Taufkirchen, Germany) to generate non-adherent neurosphere cultures.

### 2.8. Immunocytochemistry

Cells were rinsed with ice-cold PBS, fixed in 4% paraformaldehyde in PBS for 30 min at 4 °C, washed three times with PBS, blocked with 5% BSA, permeabilized with 0.1% Triton X-100 for 10 min at 4 °C, and washed three times in PBS. The cells were then incubated with rabbit anti-Tpr (1:1000; Abcam, AB84516), rabbit anti-phospho-Tpr (1:1000, S-2059, kind gift of V.K. Nandicoori [47]), mouse anti-FG-Nups (1:1000; Abcam, AB24609), and mouse anti-Cdk1 (1:500; Invitrogen, MA5-15824) in PBS with 1% BSA overnight. After three washes in PBS, the cells were incubated with a secondary antibody conjugated with Alexa Fluor 488 or 594 (1:200; Jackson ImmunoResearch Laboratories) for 1 h in PBS with 1% BSA at room temperature, washed three times in PBS, and cover-slipped with DAPI (Southern Biotechnology).

### 2.9. Tpr Knockdown Experiments

For Tpr knockdown experiments, wild-type NSPCs were transfected by electroporation with 250 nM of Tpr siRNA (Horizon Discovery, Cambridge, UK, L-041152-01-0005) or control siRNA (Horizon Discovery, D-001830-20-05) using the Mouse Neural Stem Cell Nucleofector Kit (Lonza, Basel, Switzerland, VPG-1004), following the manufacturer’s instructions. Electroporated cells were cultured for 1 day.

### 2.10. Immunoblots

To detect Tpr and Nup153 expression in adult NSPCs, primary cells were cultured under proliferation (+rhFGF2, +rhEGF) and differentiation conditions (w/o rhFGF2 and rhEGF) for 2 days. Cells were lysed in a cell lysis buffer (1% Nonidet P-40, 150 mM NaCl, 1 mM EDTA, and 20 mM Tris HCl, pH 8.0) supplemented with protease/phosphatase inhibitor cocktails (Calbiochem, Darmstadt, Germany). For co-immunoprecipitation, 1 mg of NSPC cell lysate was prepared in the cell lysis buffer and incubated with rabbit anti-Tpr antibody (5 µg, Abcam, AB70610) bound to protein A Sepharose beads for 4 h at 4 °C. After three washes, the beads were resuspended in sample buffer, boiled for 10 min, and centrifuged. Protein extracts from cell culture and co-immunoprecipitation experiments were separated by electrophoresis on 4–12% SDS-PAGE gels [53] and detected using the following antibodies: rabbit anti-Tpr (1:1000; Abcam, AB70610,), rat anti-Nup153 (1:1000; Santa Cruz, sc101544,), rabbit anti-Phospho-Tpr (1:1000, S-2059, kind gift of V.K. Nandicoori [47]), and rabbit anti-GAPDH (1:1000; Cell Signaling, Leiden, The Netherlands 2118). Blots were washed three times with TBS-T, incubated with peroxidase-labeled secondary antibodies (goat anti-rabbit IgG, Cell Signaling Technology, 1:5000; goat anti-rat IgG, 1:5000, Cell Signaling Technology; goat anti-mouse IgG, Santa Cruz Biotechnology), Heidelberg, Germany, diluted in 5% nonfat milk in TBS-T for 1 h at room temperature, and washed again, followed by detection with chemiluminescence (ECL, GE Healthcare, Bruchsal, Germany).

### 2.11. RNA Isolation and Quantitative PCR

RNA was isolated from primary NSPCs, and quantitative real-time PCR was performed as described previously [46]. The following primers were used:
*Cdk1*:Fwd 5′-AGA AGG TAC TTA CGG TGT GGT-3′
Rev 5′-GAG AGA TTT CCC GAA TTG CAG T-3′*GAPDH*:Fwd 5′-CAAGGCCGAGAATGGGA-3′
Rev 5′-GGCCTCACCCCATTTGAT-3′

### 2.12. Mass Spectrometry

The interaction partners for Tpr and P-Tpr were determined by mass spectrometry and the S-Trap method. Co-immunoprecipitation samples were further processed by the S-Trap method [54]. One µg of peptides was analyzed on a Q-Exactive Plus mass spectrometer (Thermo Fisher Scientific, San Jose, CA, USA) coupled to an EASY-nLCTM 1000 UHPLC system (Thermo Fisher Scientific), and quantification was performed with the MaxQuant technique [55]. For label-free quantification, both the MaxLFQ algorithm and the iBAQ (Intensity Based Absolute Quantification) algorithm were applied with the standard settings. To calculate the log2 fold-change for every protein, intensities of proteins not detected in one condition (intensity = 0) were corrected to 10,000. Only unique peptides were used for quantification, and the protein IDs were assigned according to the mouse-EBI-reference database (https://www.ebi.ac.uk).

### 2.13. Fluorescence Microscopy Imaging

A Leica TCS SP8 confocal microscope with 20×, 40×, and 63× oil immersion objectives with 5× zoom (when acquiring single cells) was used for the z-stack acquisition of cells and tissues for 3-D reconstruction. The z-stack size was set to 0.5 or 1.0 µm for immunohistochemistry and immunocytochemistry, depending on the quantification method. For the quantification of total expression level, immunoreactivity was calculated with LAS AF software. The polygon tool was used to manually define the cell shape in every z-stack, with DAPI as a reference for the cell volume and the antibody DCX to label neuroblast cells. Tpr expression was calculated using the total pixels per area in different z-stacks, then averaging the pixels after subtracting noise measured from tissue regions where there were no cells. For Tpr expression at the NPC, another inner polygon was created at the inner nuclear membrane and subtracted from the outer nuclear membrane for overall Tpr expression in different NSPC types.

### 2.14. Ethical Statement

All subjects gave their informed consent for inclusion before they participated in the study. The study was conducted in accordance with the Declaration of Helsinki (2013), and the protocol was approved by the local Ethics Committee of the Albert-Ludwigs-University Freiburg, Germany (authorization: 10008/09, approved 19 June 2009). All animal experiments were approved by the Federal Ministry for Nature, Environment, and Consumer Protection of the state of Baden-Württemberg and were performed in accordance with the respective national, federal, and institutional regulations (authorizations: X-17-01A (approved 26 January 2017), X-20-03C (approved 20 February 2020), and G 13-093 (approved 18 December 2013).

### 2.15. Statistical Analysis

Data are shown as the means ± SEM. For multiple comparisons, differences between groups were assessed by one-way ANOVA followed by Bonferroni’s correction for comparison of means. To assess the significance of differences between the two groups, unpaired two-tailed or one-tailed Student’s *t*-tests were used. All statistical analyses were conducted in GraphPad Prism (GraphPad Software, v6).

## 3. Results

### 3.1. Phosphorylation Determines the Subcellular Localization of Tpr in NSPCs

To determine the role of Tpr in adult NSPCs, we first examined its detailed subcellular localization in the NSPC lineage. We used SR Airyscan microscopy to identify the subcellular localization of Tpr in individual SVZ-derived NSPCs (Figure 1A). NPC imaging is traditionally carried out with an antibody to the FG-Nups to detect mainly Nup62, Nup214, and Nup358, which are located in the center and at the cytoplasmic side of the NPC [56], with a minor signal from the nuclear basket protein Nup153. Immunolabeling for Tpr, which is located on the nucleoplasmic side of the NPC [30], in combination with immunolabeling for FG-Nups, revealed distinct localizations of Tpr and FG-Nups within the NPC in SVZ-derived NSPCs, with the Tpr signal closest to the nucleus (Figure 1B–E).

In these experiments, the SR Airyscan can resolve subtle differences in epitope locations within the NPC in NSPCs (NPC external diameter ~120 nm and ~25 nm for the nuclear ring moiety). By contrast, recorded 3-D image stacks of cells from the same sample with state-of-the-art confocal laser scanning microscopy (CLSM) and applied deconvolution did not resolve the individual Nups within the NPCs of NSPCs (Figure 1B, left). Surprisingly, unlike Tpr, immunolabeling for P-Tpr revealed a nuclear localization in SVZ-derived NSPCs (Figure 1F and Appendix A), suggesting different functions for Tpr and P-Tpr in NSPCs.

Next, we analyzed Tpr expression in the SGZ stem cell niche of the hippocampus. The lineage progression of NSPCs in the SGZ can be distinguished by differential marker expression [57]. Proliferating Ki-67+ progenitor cells and doublecortin (DCX)+ immature, newborn neurons within the SGZ had nuclear P-Tpr expression, whereas differentiated, mature NeuN+ neurons of the GCL had P-Tpr expression only at the nuclear envelope (Figure 1G). As expected, Tpr was located on the nucleoplasmic side of the NPCs in all examined NSPCs (Appendix A). Importantly, electron microscopy confirmed nuclear localization of P-Tpr in NSPCs of the SGZ, whereas the P-Tpr signal was absent from the nucleus of mature interneurons of the hippocampus (Figure 1H and Appendix A). As expected, the FG-Nups and Tpr signals were located throughout the inner conduit of the NPC and in the nuclear basket area, respectively (Figure 1H and Appendix A). Overall, these results show that P-Tpr is located in the nucleus of newborn neuroblasts of the SGZ in the hippocampus, whereas Tpr is located on the nucleoplasmic side of NPCs in NSPCs.

### 3.2. Tpr Levels Increase Prior to Hippocampal Neurogenesis in 5xFAD Mice

NPC components can become misregulated in neurons in AD [18,19]. However, how Tpr expression and localization changes in NSPCs during AD is unknown. We first sought to determine the Tpr expression in proliferating and differentiating SVZ-derived NSPCs isolated from wild-type (WT) mice in vitro, as Nup153 expression levels were recently described to decrease with NSPC differentiation (Figure 2A, left, [41]). Surprisingly, Tpr expression levels increased with NSPC differentiation (Figure 2A, right), opposing the described Nup153 expression in NSPCs. 

Next, we evaluated changes in Tpr expression in the NSPC lineage in WT mice. Newborn neuroblasts in the SGZ of the hippocampus (DCX+) had lower Tpr levels than mature neurons in the GCL of the hippocampus (Figure 2B). We then analyzed Tpr expression in detail via individual double immunolabeling of cells in the SGZ NSPC lineage from proliferating NSPCs (Ki-67+), to neuroblasts (DCX+), to mature neurons (NeuN+) (Figure 2C), to confirm the increase in Tpr expression during neuronal differentiation in SGZ NSPCs. We found that the Tpr levels were fourfold higher in neuroblasts and sevenfold higher in mature neurons compared with proliferating NSPCs of the SGZ in the hippocampus (Figure 2D,E). The same pattern of Tpr expression was observed in the SVZ NSPC neurogenic lineage (Appendix A), and electron microscopy revealed lower Tpr expression levels in SGZ NSPCs than in mature hippocampal interneurons (Appendix A). Interestingly, P-Tpr levels, like Tpr levels, increased from precursor cells to mature neurons in the hippocampal SGZ, but P-Tpr localization changed in the neurogenic lineage from a nuclear pattern in proliferating NSPCs to a nuclear-envelope pattern in mature neurons (Appendix A).

Next, we evaluated changes in Tpr expression in NSPCs in two mouse models of AD. First, we compared 5xFAD transgenic mice at different ages (6 weeks, 4 months, and 8 months) to their respective WT littermate controls. 5xFAD transgenic mice show amyloid-β (Aβ) deposition with 4 months of age [44,58] and early expression of hippocampal amyloid precursor protein (APP) (Appendix A). In a published study, 5xFAD mice had an increase in the number of Ki67+ proliferating NSPCs, as well as increased neurogenesis and more DCX+ immature granule cells at 4 months of age (Figure 2F and [44]) but a significant decrease in neurogenesis at 8 months of age. Thus, these mice are ideal for examining whether Tpr expression is correlating with neurogenesis. Importantly, Tpr expression in 5xFAD mice was increased in DCX+ neuroblasts at 6 weeks of age, well before the increase in hippocampal neurogenesis at 4 months of age, and Tpr expression was decreased in DCX+ neuroblasts at 4 months of age, well before the decline in neurogenesis (Figure 2G,H, [44]). Tpr expression did not change significantly in hippocampal Ki67+ NSPCs or NeuN+ neurons in 6-week-old or 4- and 8-month-old 5xFAD mice (Appendix A). Next, in addition to the 5xFAD transgenic mice, we analyzed Tpr expression in the APP23 mouse model of AD. We found decreased Tpr expression in hippocampal DCX+ neuroblasts that correlated with increased neurogenesis in 9-month-old APP23 mice (Appendix A). This is the time when the mice just begin to develop amyloid plaques [45]. Overall, our data reveal an early misregulation of Tpr expression in DCX+ neuroblasts in the two mouse models of AD. This misregulation correlates with changes in neurogenesis in the models, suggesting that misregulated Tpr in AD associates with NSPC function and differentiation.

### 3.3. Tpr and P-Tpr Interactome in NSPCs

We next sought to understand the functional relevance of the increase in Tpr expression in NSPCs along the neurogenic lineage, the different localization of Tpr and P-Tpr in NSPCs, and the aberrant Tpr expression in 5xFAD and APP23 mice. We combined endogenous immunoprecipitation (IP) with NSPC lysates and Tpr or P-Tpr as bait (Appendix A) and mass spectrometry (MS). We identified 759 Tpr-interacting proteins and 119 P-Tpr-interacting proteins (log2 FC ≥ 1.0 was set as the cut off) (Figure 3A,B). As expected from its nuclear basket localization, Tpr interacts with different Nups, including the known Tpr interaction partner Nup153 and also Nup50, Nup133, and Nup155 (Figure 3C) [59] (Figure 1A). In stark contrast, P-Tpr did not interact with the Nups (Figure 3C), confirming its nuclear localization in NSPCs (Figure 1F,H).

Our interactome analysis, classified via the Gene Ontology nomenclature, identified common major biological categories for Tpr and P-Tpr interacting proteins, such as ‘enzymatic activity’, ‘DNA, RNA and nucleotides’, ‘protein binding’, and ‘transcription’ D and Appendix A. Individual Nups, in particular Tpr as the nuclear basket protein, have been implicated in numerous processes beyond protein import and export, including chromatin remodeling, control of gene expression and protein modification, as well as mRNA processing and export [32,33,34,35,36,37]. As expected, Tpr interacted with several chromatin modifiers involved in nucleosome sliding and chromatin looping, such as members of the Polycomb repressor complex (PRC1) [36] and the SWI/SNF complex [60,61], that are important for NSPC maintenance and differentiation. Importantly, our MS analysis revealed different interacting chromatin modifiers for Tpr and P-Tpr. Tpr interacted with BAF complex members (Actl6a, Brg1, BAF170), the SWI/SNF complex member Arid1a, and the PRC1 complex members CBX2 and RuvBl1 (Figure 3E,F, Appendix A). P-Tpr interacted with proteins involved in control of gene expression and chromatin reorganization (Hist1h2af, Hmgb2; Figure 3F and Appendix A, Appendix A).

Interestingly, Tpr interacted with a range of transcription factors important for NSPC maintenance and differentiation, such as Sox4, Eny2, Otx1, Nfix, Hoxa5, and Smads, as well as Sox2 (Figure 3F, Appendix A), which interacts and cooperates with Nup153 to regulate the cellular state of neural progenitor cells [41]. P-Tpr interacted with Bclaf1, Tpt1, and Btf3, which have largely unknown functions in NSPC fate regulation (Figure 3F, Appendix A).

Finally, P-Tpr interacted uniquely with Cdk1 (Figure 3F and Appendix A, Appendix A), which drives mitotic entry and progression, suggesting a role for P-Tpr in mitosis of NSPCs. Cdk1 phosphorylates multiple epigenetic regulators, and its substrates are chromatin-bound [62]. As expected, P-Tpr interacted uniquely with other proteins in NSPCs involved in mitosis and cell-cycle regulation (Cc2d1b, Tubb2b) as well as in metabolism (Rag1, Glyr1, Mri1, Atp6v0c) and cytoskeletal organization (Cask) (Appendix A, Appendix A), suggesting unique functions for P-Tpr in cycling NSPCs. Surprisingly, electron micrography revealed that P-Tpr localized mainly to heterochromatin in the NSPC nucleus of the hippocampal SGZ (Figure 3G). Cdk1 regulates genomic stability in stem cells [63], and Cdk1 phosphorylates Tpr to determine its distinct chromatin localization [47]. Thus, P-Tpr–Cdk1 together might regulate genome organization in SVZ-derived and hippocampal NSPCs; the abundance and localization of this complex may be misregulated in AD, potentially affecting Tpr localization and neurogenesis. Indeed, Cdk1 expression is reduced in hippocampal tissue from 8-month-old 5xFAD mice (Appendix A). In summary, these results demonstrate common and unique Tpr and P-Tpr interaction partners in NSPCs and suggest that, in mouse models of AD, aberrant expression and localization of Tpr and P-Tpr in NSPCs of the SVZ and hippocampal SGZ regulate essential NSPC functions.

### 3.4. Aberrant Tpr Expression Correlates with Altered NPC Counts in NSPCs in a Mouse Model of AD

Tpr regulates the NPC count in various cell types [34], which is critical for cell differentiation [40]. However, does misregulation of Tpr affect the NPC count and NSPC differentiation in AD? NPC counts for individual cells in tissue sections have never been conducted. Using Imaris 3-D processing of SR Airyscan images, we set up a novel quantification method to analyze NPC counts per cell volume for individual Tpr immunolabeled hippocampal NSPCs (Figure 4A). We confirmed our NPC analysis on primary SVZ-derived NSPCs. Decreased Tpr expression results in increased NPC counts in various cell types [34]. In agreement with this result, Tpr depletion by siRNA in primary SVZ-derived NSPCs resulted in increased NPC counts (Figure 4B and Appendix A). Excitingly, the NPC count decreased in hippocampal NSPCs in 5xFAD mice at 6 weeks of age (Figure 4C). This matches the timeline of the increased Tpr expression in this mouse model of AD (Figure 2G) and preceded the increased differentiation of hippocampal DCX+ neuroblasts, which occurs at 4 months of age (Figure 4F, [44]).

A direct link between the NPC and AD was established recently by showing that pathological tau can disturb the functional integrity of NPCs in neurons [18]. However, how Tpr expression changes in the hippocampal SGZ of human AD individuals is unknown. Tpr expression levels in the hippocampal SGZ of AD individuals were lower than in healthy age-control samples (Figure 4D), suggesting that misregulated Tpr expression is involved in loss of NPC integrity and might affect human hippocampal NSPC differentiation in AD. Overall, our results suggest that misregulated Tpr expression controls hippocampal NSPC NPC counts and architecture in AD.

## 4. Discussion

Adult neurogenesis is critically involved in learning and memory and appears to be altered in neurodegenerative diseases, such as AD. NPCs are critically maintaining the equilibrium between the nucleus and cytoplasm and aberrant NPC functioning has been implicated in neurodegeneration. However, how the organization of the individual components of the NPC changes in NSPCs in AD, and how this might affect adult neurogenesis are fundamental and unresolved questions. Our results suggest that the dynamic expression and localization of Tpr regulates hippocampal NSPC NPC architecture in mouse models of AD. Our results suggest that the localization of the nucleoporin Tpr in subcellular structures determines its interaction partners and that Tpr levels determine the NPC count in hippocampal NSPCs. Our results also suggest that Tpr misregulation is associated with impaired neurogenesis in AD. 

Our results are consistent with the following working model—(i) Nucleoporin Tpr is localized to the nuclear basket in NSPCs, and P-Tpr is localized to the nucleus. (ii) Tpr expression increases throughout neurogenesis with the highest expression in postmitotic SVZ and hippocampal neurons. (iii) Aberrant hippocampal NSPC Tpr expression precedes altered hippocampal neurogenesis in mouse models of AD. (iv) Tpr and P-Tpr interact with common and unique proteins in NSPCs, and (v) aberrant Tpr expression levels define the NPC count in AD. Thus, the dynamic expression and localization of Tpr may regulate impaired hippocampal neurogenesis in AD.

The NPC composition is not uniform across tissues and the insertion of particular Nups is required for differentiation along a specific lineage or for tissue homeostasis [64]. Furthermore, individual Nups regulate the cell identity of embryonic [38,39,40] and adult NSPCs [41]. Interestingly, the expression level of individual Nups change among different cell types, e.g., Nup153 expression levels are high in precursor cells and decline with differentiation [40,41]. Our data suggests that Tpr expression level correlates with the degree of the cellular plasticity. While the nuclear basket protein Nup153 is highly expressed in NSPCs and is selectively downregulated in neurons, our data for Tpr reveal lower expression in NSPCs that increases with further differentiation into neurons, suggesting unique functions for Tpr in hippocampal NSPC differentiation towards newborn neurons. While Nup153 interacts with SOX2 to co-regulate genes and maintains neural progenitor cells [41], our data showed that Tpr regulates the NPC number, interacts with chromatin modifier and with unique TFs in NSPCs. Future studies will determine common and individual functions of the individual Nups, such as for Nup153 and Tpr, to better understand their roles in NPC assembly, chromatin interaction, and organization, as well as their modulation of gene expression, to potentially orchestrate adult NSPC neurogenesis.

Phosphorylation of Nups transduces cellular signals into crucial functions such as nucleocytoplasmic transport and cell-cycle progression [65]. Tpr is crucial for progression of mitosis across different species [66,67,68], and P-Tpr was shown to have differential localization to chromatin during mitosis in HeLa cells [47]. Our data demonstrate P-Tpr nuclear localization in hippocampal DCX+ neuroblasts, but not in postmitotic neurons, suggesting a function for P-Tpr in mitosis of hippocampal NSPCs. We identified Cdk1, which is the master regulator of the cell-cycle progression [69], as P-Tpr interaction partner in NSPCs. Cdk1 regulates genomic stability in stem cells [63], and Cdk1 phosphorylates Tpr to determine its distinct chromatin localization [47]. Thus, we suggest that P-Tpr–Cdk1 together might regulate genome organization during cell division in SVZ-derived and hippocampal NSPCs. Determining the underlying mechanism of Tpr phosphorylation and the P-Tpr–Cdk1 co-regulatory role in chromatin organization and gene regulation will be crucial for understanding how phosphorylation of individual Nups, such as Tpr phosphorylation, regulate NSPC fate.

Impairment of adult hippocampal neurogenesis is an early event in AD [8] and might play a crucial role in the cognitive dysfunction associated with AD pathology [9,10]. In AD, irregularities in nuclear morphology suggest a contribution of nuclear dysfunction to pathophysiology [19], and tau interacts with NPC components, leading to their mislocalization and subsequent disruption of NPC function in neurons [18,70]. Beside the described functional decline of NPCs in transport related roles in neurons, a role for individual Nups in NSPCs in neurodegenerative diseases, in particular in adult hippocampal neurogenesis in AD, is only poorly described. Alterations of Nup153 level has implications in the maintenance and differentiation of cultured hippocampal neural stem cells in a mouse AD model. Tpr expression in NSPCs in AD has yet not been addressed. While Nup153 expression is reduced in these NSPCs of AD mice [43], our data, in stark contrast, shows that Tpr expression is increased in hippocampal NSPCs during the early stages in the 5xFAD mouse model of AD, before the onset of Aβ pathology and neuronal loss. Furthermore, the increased Tpr expression preceded the neurogenic response in 5xFAD mice and regulated NSPC features, such as their NPC count. We also confirmed the altered Tpr expression pattern in APP23 mice, another mouse model of AD. Given the important functions of Tpr in regulating NPC numbers and its interaction with specific chromatin modifiers and TFs in NSPCs (our study), early changes in Tpr expression might alter NSPC fate in AD. The mechanisms that regulate Tpr expression in hippocampal NSPCs in AD are currently unknown, and are the focus of future studies to potentially modulate Tpr expression and neurogenesis in AD in the future. 

While both AD models in our study, 5xFAD and APP23, are associated with an altered neurogenesis prior to disease pathology, we showed that increased Tpr expression in DCX+ hippocampal neuroblasts precedes increased neurogenesis and that misregulation of Tpr in NSPCs correlates with altered NPC numbers in 5xFAD mice. NPC numbers change throughout differentiation, such as for neural progenitor cells, which give rise to multiple neuronal cell types with significantly different NPC numbers [41,71] and the number of NPC per cell are believed to be critical for cell differentiation [40]. Our initial analysis of human AD individuals showed a reduced Tpr expression compared to healthy control individuals. As a next step, it will be critical to understand, if early Tpr misregulation and altered NPC numbers in NSPCs in the hippocampal SGZ might contribute to a decline in neurogenesis in AD. 

NSPC differentiation is often accompanied by large-scale nuclear reorganization, resulting in global gene expression changes. Accumulating evidence suggests that NPC components are pivotal in cell-type-specific gene regulation. Our data reveal that Tpr and P-Tpr interact with unique TFs and chromatin modifier, suggesting distinct functions for Tpr and P-Tpr in NSPCs. The nuclear basket provides anchors for mRNA export and interacts directly with the transcriptional regulatory machinery [32,72,73]. This includes the highly conserved transcription elongation and RNA export (TREX-2) complex that associates stably with the NPC, and this association requires, among other nuclear basket proteins, Tpr [37,74]. The Tpr interactome (our data) included interaction partners belonging to the TREX-2 complex (e.g., ENY2 and PCID2), suggesting a role for Tpr in mRNA export in NSPCs. P-Tpr is potentially involved in regulating cell-cycle progression, as P-Tpr interacts with Cdk1. Furthermore, our interactome results revealed unique chromatin interaction partners for Tpr and P-Tpr in adults NSPCs, suggesting their different function in chromatin organization. While our study revealed different interactomes for Tpr and P-Tpr in NSPCs, further high-throughput studies are necessary to describe how altered posttranslational modifications of NPC nucleoporins affect their interactome and functional outcome in regulating NSPC maintenance and differentiation.

Individual Nups can become mislocalized or altered in expression in neurodegeneration [17,18,19,20,21]. However, the underlying mechanisms by which Nup expression is altered remains elusive. In AD, tau aggregation has been shown to cause NPC invagination and defective mRNA export [75], and tau interacts with Nup98 leading to its mislocalization from NPCs to the cytosol [18]. Our data indicate early changes in Tpr expression in NSPCs in two mouse model of AD, before any pathological amyloid-β deposition, suggesting that altered Tpr expression is an early event in disease pathogenesis. We speculate that the altered expression of Tpr and its nuclear localization in proliferating NSPCs upon phosphorylation might trigger the events leading to aberrant neurogenesis in AD. How Tpr is regulated in NSPCs in AD remains unknown.

Oxidative/nitrosative stress leads to reduced Nup153 expression and disturbed differentiation of NSPCs in vitro [43], suggesting that early mitochondrial dysfunction in AD might contribute to early misregulation of Nups in AD. In ALS, decreased nuclear export of the ESCRT-III protein CHMP7 is associated with pathogenic reduction of Nups, including Tpr, suggesting that proteasomal and lysosomal degradation pathways are involved in Nup homeostasis and related to neurodegeneration [76]. Yet, whether Tpr misregulation is a consequence of pathological protein aggregation in AD or rather contributes to the development is unknown and future experiments will address the underlying mechanism of Tpr regulation in NSPCs in AD.

In this study, we made the first effort to identify the individual roles of Tpr and P-Tpr in NSPC differentiation in AD. Tpr and P-Tpr had different localization patterns in NSPCs and unique interaction partners, and their expression is highly regulated in AD (our study) in the opposite direction to the regulation of Nup153 [41]. Identification of the regulators of adult hippocampal neurogenesis under pathological conditions [77] is key to pinpointing targets for stem cell therapy. Future in vivo studies with NSPC-specific deletion of Tpr will be required to validate and further describe the functions of Tpr in regulating the neurogenic NSPC program in AD. Our study suggests that manipulation of AD-induced misregulated Tpr expression and function in human hippocampal neural stem cells (e.g., by tight and timely regulation of Tpr expression) may control hippocampal NSPC fate to boost neurogenesis and counteract cognitive decline.

## Figures and Tables

**Figure 1 cells-12-02757-f001:**
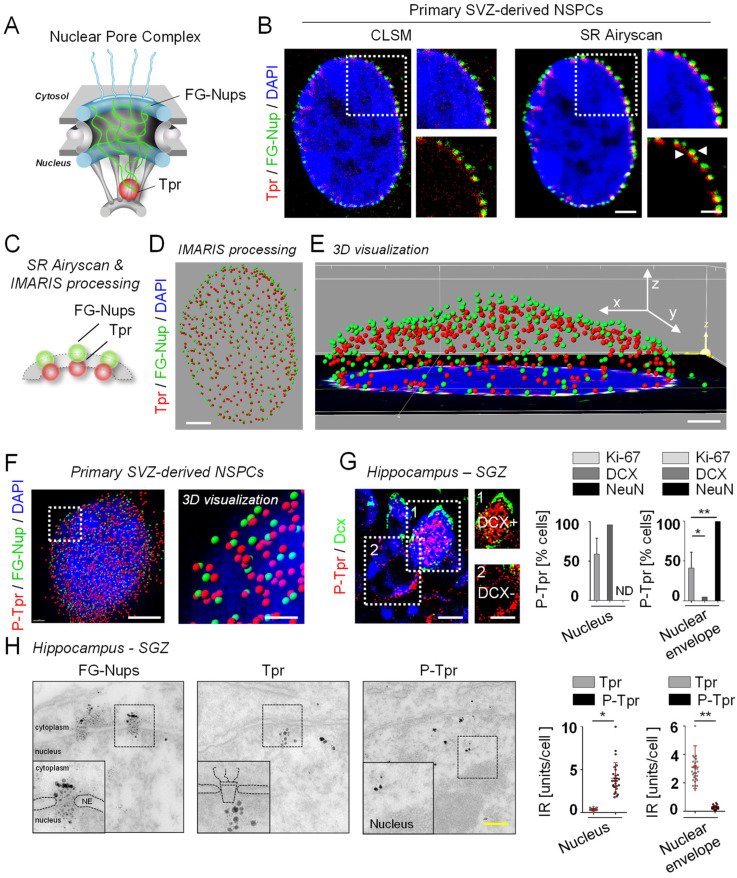
Translocated promoter region (Tpr) phosphorylation determines subcellular localization in neural stem/precursor cells (NSPCs). (**A**) Scheme illustrating the localization of Tpr and phenylalanine-glycine repeat nucleoporins (FG-Nups) in nuclear pore complexes (NPCs). (**B**) Comparative imaging of NPCs in subventricular zone- (SVZ-) derived NSPCs immunolabeled with Tpr (red) and FG-Nups (green) using confocal laser scanning microscopy (CLSM) (left) and SR Airyscan microscopy (right). Enlargements at the right of each panel indicate representative resolutions of the Tpr and FG-Nup signals, and white arrowheads indicate the distinct localizations of TPR and FG-Nups resolved with SR Airyscan microscopy. Nuclei are stained with DAPI (blue). Scale bars: 2 µm, 500 nm (enlargement). Representative images from three independent experiments (10 nuclei were analyzed in total). (**C**) Scheme illustrating Tpr localization relative to FG-Nups in an NPC after SR Airyscan microscopy and Imaris 3-D processing. (**D**) Representative SR Airyscan microscopy and Imaris-processed image of an SVZ-derived NSPC nucleus immunolabeled for Tpr (red) and FG-Nups (green). Representative image from three independent experiments (10 nuclei were analyzed in total). Scale bar: 2 µm. (**E**) 3-D visualization revealing distinct localizations for Tpr (red) and FG-Nups (green) in the NPC of SVZ-derived NSPCs. Representative image from three independent experiments (10 nuclei were analyzed in total). Scale bar: 1 µm. (**F**) 3-D visualization of confocal images revealing nuclear phospho-Tpr (P-Tpr, red) localization in NSPCs in vitro. Representative images from three independent experiments (10 nuclei were analyzed in total). Scale bars: 2 µm, 500 nm (enlargement). (**G**) Immunolabeling for P-Tpr (red) in combination with doublecortin (DCX) (green, marker for neuroblasts) in the subgranular zone (SGZ) of the hippocampus in adult wild-type (WT) mice. Enlargements indicate a DCX+ cell with nuclear P-Tpr and a DCX−cell with no nuclear P-Tpr. Quantification of nuclear or nuclear envelope P-Tpr localizations in Ki-67+, DCX+, and NeuN+ cells in the SGZ of the hippocampus in adult WT mice (*n* = 4 mice, 79–98 single cells were analyzed per Ki-67, DCX, or NeuN condition). Scale bars: 3 µm, 5 µm (enlargement). Values are the mean ± SEM (*p*-values calculated by one-way ANOVA with Bonferroni’s multiple comparisons test, * *p* < 0.05, ** *p* < 0.01) (**H**) Electron microscopy determining the localization of Tpr and P-Tpr in comparison to FG-Nups in NSPCs of the hippocampal SGZ in adult WT mice. Quantification of nuclear or nuclear envelope Tpr- and P-Tpr-IR in NSPCs of the hippocampal SGZ in adult WT mice (*n* = 3 mice, 27–30 single cells were analyzed per Tpr or P-Tpr localization). Scale bar: 100 nm. Values are the mean ± SEM (*p*-values calculated by unpaired Student’s *t*-test, * *p* < 0.05, ** *p* < 0.01).

**Figure 2 cells-12-02757-f002:**
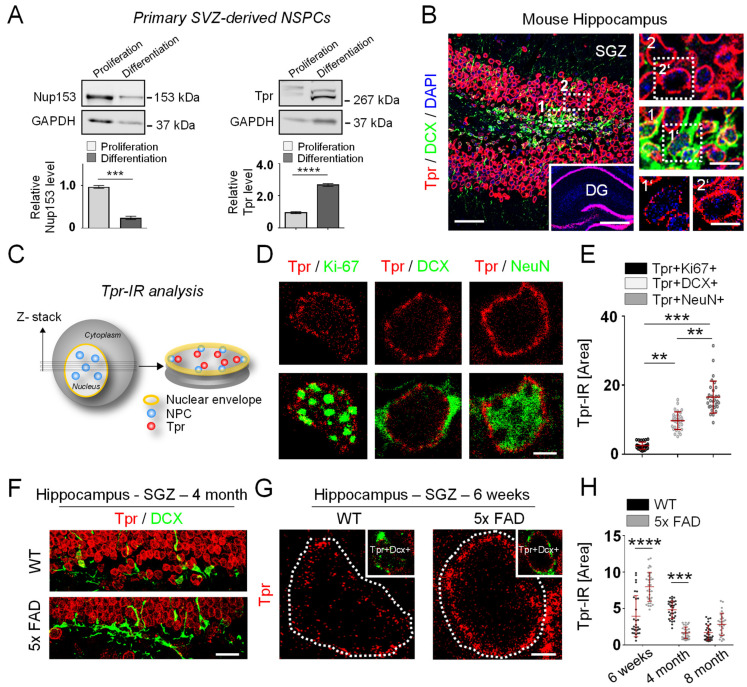
Increased Tpr expression precedes hippocampal neurogenesis in 5xFAD mice. (**A**) Protein expression of Tpr (right) compared to Nup153 (left) in proliferating and differentiating NSPCs determined by western blotting. Quantification graphs of the Nup153 protein level (*n* = 3) and Tpr protein level (*n* = 4) are shown. Blots were quantitated by densitometry and normalized to GAPDH. Values are the mean ± SEM (*p*-values calculated by unpaired student’s *t*-test, *** *p* < 0.001, **** *p* < 0.0001). (**B**) Immunolabeling for Tpr (red) and DCX (green) in the hippocampal SGZ in adult WT mice Inset: representative overview image of the hippocampus immunolabeled for Tpr (red). Representative image from a total of 10 different WT mouse sections analyzed. Enlargements indicate a DCX+ cell with weak Tpr immunoreactivity (1, 1′) compared to a DCX- cell with strong Tpr immunoreactivity (2, 2′). Scale bars: 50 µm, 500 µm (inset), 10 µm (enlargements, 1, 1′, 2, 2′). (**C**) Scheme illustrating Tpr levels in nuclei of individual cells in the hippocampus of adult WT mice. (**D**) Immunolabeling for Tpr (red) and Ki-67, DCX, and NeuN (all green) in individual cells in the hippocampal SGZ and the molecular layer of adult WT mice. Scale bar: 5 µm. (**E**) Quantification of Tpr immunoreactivity in Ki-67+, DCX+ and NeuN+ cells in the hippocampal SGZ and granule cell layer of adult WT mice (*n* = 3 mice, 28–32 single cells were analyzed per Tpr+Ki-67+, Tpr+DCX+, or Tpr+NeuN+ condition). Values are the mean ± SEM (*p*-values calculated by one-way ANOVA with Bonferroni’s multiple comparisons test, ** *p* < 0.01, *** *p* < 0.001). (**F**) Immunolabeling for DCX (green) and Tpr (red) in the hippocampus of 5xFAD and WT mice sacrificed at 4 months of age. Scale bar: 50 µm. (**G**) Immunolabeling for Tpr (red) in the SGZ of the hippocampus of 5xFAD and WT mice sacrificed at 6 weeks of age. Inset: only Tpr+DCX+ cells in the SGZ of the hippocampus were analyzed. Scale bar: 2 µm. (**H**) Quantification of Tpr immunoreactivity in DCX+ cells in the hippocampal SGZ of 5xFAD and WT mice at 6 weeks and 4 and 8 months of age (*n* = 3 mice per condition, 29–34 single cells were analyzed per time point for WT and 5XFAD mice). Values are the mean ± SEM (*p*-values calculated by one-way ANOVA with Bonferroni’s multiple comparisons test, *** *p* < 0.001, **** *p* < 0.0001).

**Figure 3 cells-12-02757-f003:**
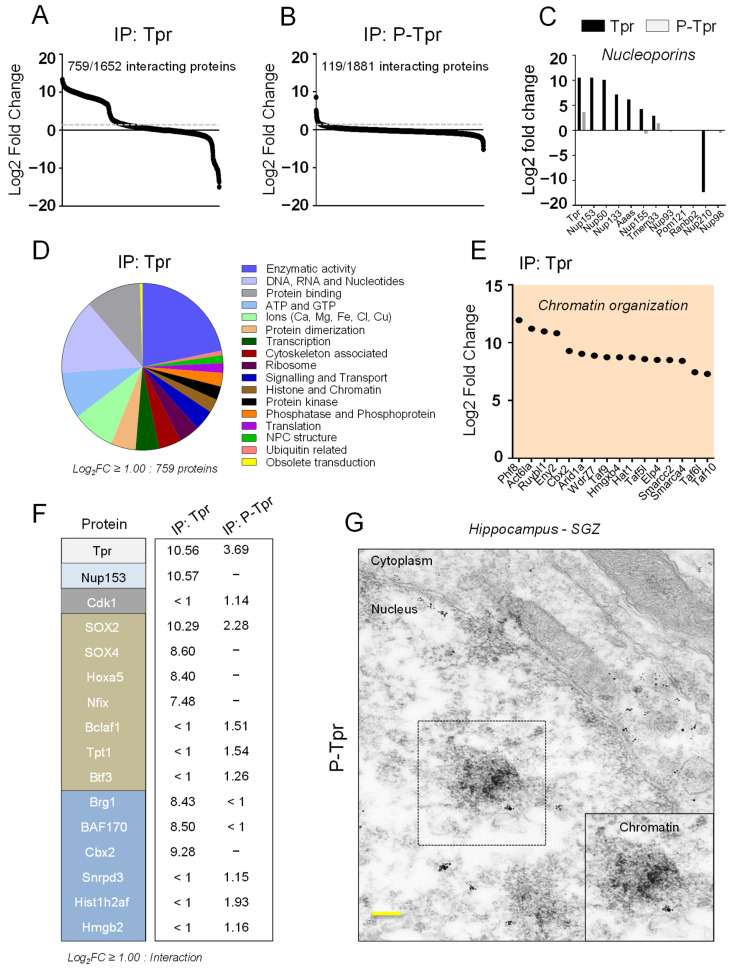
Tpr and P-Tpr interactome in NSPCs. (**A**,**B**) Plots showing Tpr and P-Tpr interaction partners in SVZ-derived NSPCs identified by Tpr and P-Tpr immunoprecipitation and mass spectrometry. The dotted grey line indicating a log2 fold-change value ≥ 1.0 was set as the threshold for Tpr and P-Tpr protein interactions. (**C**) Nucleoporin interaction partners of Tpr and P-Tpr. Nup153 served as a positive control for Tpr interaction. (**D**) Pie chart of identified Tpr interaction partners in NSPCs classified via the Gene Ontology nomenclature. (**E**) Plot showing the top 16 interaction partners with a log2 fold value ≥ 5.0 for Tpr involved in chromatin organization and control of stem cell fate. (**F**) Comparison of Tpr and P-Tpr interaction partners identified by mass spectrometry. (**G**) Electron microscopy images revealing P-Tpr nuclear chromatin localization of hippocampal SGZ cells in adult WT mice. Rectangular selection reveals P-Tpr localization to the heterochromatin site. Representative electron microscopy image from three independent experiments (27 single cells were analyzed in total for P-Tpr localization). Scale bar: 100 nm.

**Figure 4 cells-12-02757-f004:**
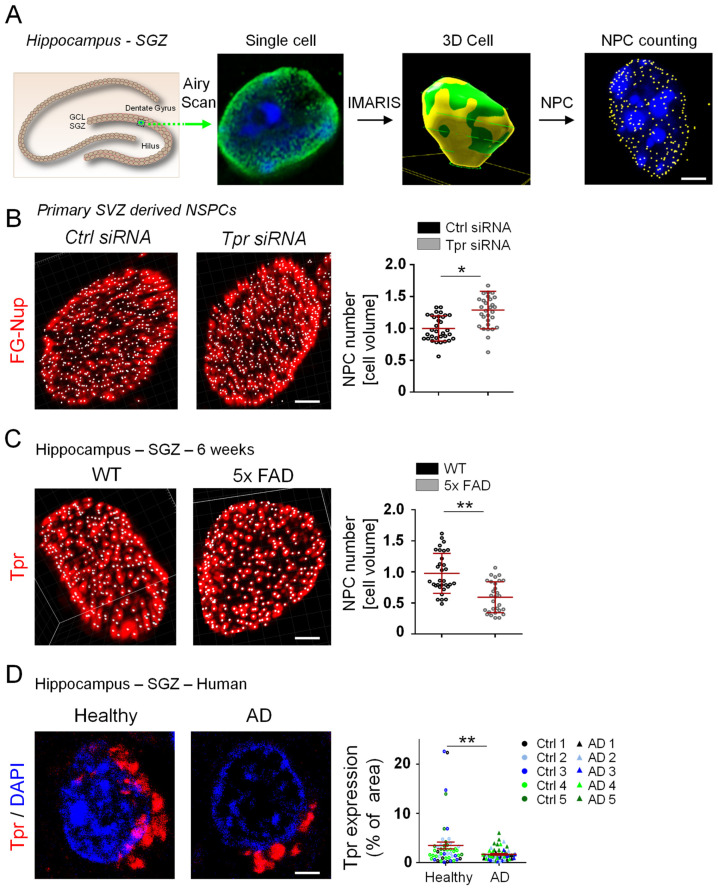
Aberrant Tpr expression correlates with altered NPC counts in hippocampal NSPCs in a mouse model of AD. (**A**) Schematic diagram illustrating the workflow of the automated NPC counting in NSPCs in hippocampal tissue sections. Scale bar: 2 µm (**B**) Immunolabeling for FG-Nups (red) in Tpr-depleted SVZ-derived NSPCs and control cells in vitro. Quantification of NPC counts in Tpr-depleted SVZ-derived NSPCs compared with controls (*n* = 3 mice, Ctrl siRNA = 30 total Nuclei and Tpr siRNA = 33 total Nuclei were analyzed). Scale bar: 2 µm. Values are the mean ± SEM (*p*-values calculated by unpaired Student’s *t*-test, * *p* < 0.05). (**C**) Immunolabeling for Tpr (red) in the SGZ of the hippocampus of 5xFAD and WT mice sacrificed at 6 weeks of age. Quantification of NPC counts in the hippocampal SGZ of 5xFAD and WT mice sacrificed at 6 weeks of age (*n* = 3 mice per condition, WT = 32 total Nuclei and 5xFAD = 29 total Nuclei were analyzed). Scale bar: 2 µm. Values are the mean ± SEM (*p*-values calculated by unpaired Student’s *t*-test, ** *p* < 0.01). (**D**) Immunolabeling for Tpr (red) in the SGZ of the human hippocampus of AD individuals compared with healthy age-matched controls. Quantification of Tpr expression in single NSPCs of the human hippocampal SGZ in AD individuals and in healthy age-matched controls (AD (AD 1–5): *n* = 5; healthy controls (Ctrl 1–5): *n* = 5, AD = 53 total single cells and healthy controls = 44 single cells were analyzed). Scale bar: 2 µm. Values are the mean ± SEM (*p*-values calculated by unpaired Student’s *t*-test, ** *p* < 0.01).

## Data Availability

The data that support the findings of this study are available from the corresponding author upon reasonable request. The mass spectrometry data reported in this paper have been deposited to GEO and are publicly available.

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
