# Peer review of "Tpr Misregulation in Hippocampal Neural Stem Cells in Mouse Models of Alzheimer’s Disease"

_cells, 2023, doi:10.3390/cells12232757_

Round 1
Reviewer 1 Report (Previous Reviewer 2)
Comments and Suggestions for Authors
The manuscript has been well-planned and written. However, some major concerns remain to be addressed.
1) The sentences and statements in the introduction need to be fine-tuned and improved.
2) In the introduction, authors can consider including the importance, and regulation/dysregulation of Tpr by various factors and pathological conditions.
3) In the methodology, the number of animals used, and age are missing though this information has been mentioned in fig legends.
4) In the results part, figures comparing the corresponding groups and different age groups appear to be inconsistent or missing.
5) In the discussion part, authors should include a segment of statements related to the expression of Tpr and associated proteins in neural stem cells and glial cells with reference to health and disease.
6) In the discussion part, a possible mechanism for the regulation of Tpr along the neurogenic program needs to be improvised regarding health, AD, and other diseases.
7) The interpretation of results, information on the existing supporting and controversial reports, novelty, significance, and drawback of this study need to be improved in the manuscript in the corresponding sections.
Comments on the Quality of English LanguageConsiderable editing of the English language is required.
Author Response
Please see the attachment.

Reviewer 2 Report (New Reviewer)
Comments and Suggestions for Authors
In this study, the authors explored the nuclear distribution of phosphorylated / unphosphorylated translocated promoter region (Tpr) in Nuclear Pore Complexes in brain of the 5xFAD or APP23 mouse models of Alzheimer’s disease (AD) or in post mortem histological brain sections of patients with AD, with a focus on the germinative, neural progenitor cell-containing, hippocampus subgranular zone (SGZ) and subventricular zone (SVZ). They also performed interactome analyses of Trp and P-Trp from neural stem/progenitor cells (NSPCs). They authors conclude that altered levels and subcellular localization of Tpr in CNS disease may affect Tpr functionality, resulting in dysfunction of NSPCs and disease.
The study is highly relevant because (1) nuclear pores play a key role in cell homeostasis and in neurodegeneration and (2) adult neuronogenesis is known to be associated with decline in cognition and memory in Alzheimer’s disease. This is an original and exciting work that deserves publication. The data sound although they sometimes lack clarification. I have the following minor requests.
METHODS :
The section « immunoblots » is confusing since it provides very few details on immunoblotting detection (ECL probably ? fluorescence ?). For the co-immunoprecipitation procedure, the sentence « Protein extracts were separated by electrophoresis on 4–12% SDS-PAGE gels [41] with the following antibodies » is also confusing : do you mean co-migration to get electrophoretic shift or, most probably, « detected with the following antibodies » ?
RESULTS
1-Phosphorylation determines the subcellular localization of Tpr in NSPCs.
Line 264 : « We used SR Airyscan microscopy to identify the sub-cellular localization of Tpr in individual NSPCs » … from which source ? (it seems from SVZ as mentioned line 270, but please clarify).
Fi. 1. B, right is very convincing. Could not it be possible to show the same SR view with P-trp immunolabeling, in complement to the more processed Fig. 1F ? please give more explanations.
Figure S1, A, right, P-trp immunolabelling : is it a Z-stack projection ? is it known why the P-Trp intranuclear immunolabelling is clustered in such foci ?
2-Tpr levels increase prior to hippocampal neurogenesis in 5xFAD mice
Line 326 : « Aberrant NPC functioning is implicated in neurodegeneration [11,46] ». I think this sentence strongly supports the design of the study and deserves a better place, ie in the introduction.
Lines 331-43 : I assume these lines refer to control mice, is it correct ? This could be indicated.
Lines 364-66 : « This misregulation [of Tpr expression] correlates with changes in neurogenesis in the models, suggesting that misregulated Tpr in AD regulates NSPC function and differentiation. »
Firstly, the neigbouring « midregulated…regulates » sounds weird. More importantly, I am not convinced that the results suggest so. They undoubtly show that « misregulated Tpr » is associated with NPSC function, but « regulates » sounds as if it was a direct, even causative, conductor. I suggest « associate » rather.
Fig 2A : I do not find it convincing regarding normalization. Fold ratio How the «fold ratio » was assessed (photometry, fluorimetry, densitometry) ? ECL is known to be poorly quantitative as compared to fluorescent detection. This issue is exacerbated by the fact that GAPDH signals with very different intensities between two lanes are shown left and right. Given that these are « Representative immunoblots from three (Nup153) and four (Tpr) independent experiments », I think it would be worth to provide a bar graph indicating error bars.
4- Misregulated Tpr expression controls hippocampal NSPC NPC counts and architecture in AD.
The last part with post-mortem human brain samples provides a strong added value, but I find the data not as strong as the corresponding text claims. Given the small number of cases (5 vs. 5) and the appearance of the graph (there are 4 "control cells" dramatically above each other), I suggest that they deserve a more careful written description: for example, "revealed" seems too strong to me, among others.
Lastly, given the low number of cases, a color code could be useful in this plot : cells from the same case should be shown with the same, case-specific, color).
Round 2
Reviewer 1 Report (Previous Reviewer 2)
Comments and Suggestions for Authors
The authors have satisfactorily addressed the clarifications and improved the quality of the manuscript.
Comments on the Quality of English LanguageNeed minor editing and improvement
This manuscript is a resubmission of an earlier submission. The following is a list of the peer review reports and author responses from that submission.
Round 1
Reviewer 1 Report
Comments and Suggestions for Authors
The authors sufficiently addressed the reviewers' concerns.
Reviewer 2 Report
Comments and Suggestions for Authors
In this study, the authors attempted to address the Trp-mediated regulation of nuclear pore complex in the hippocampal neural stem cells in AD.
This article appears innovative and interesting. The whole manuscript is well organized. However, there have been several issues and flaws that should be addressed and rectified.
1) Expansions for many abbreviations are missing at first glance (i.e. Tpr, pTpr, DCX, Cdk1).
2) The long and complex sentences need to be simplified in the overall text.
3) The manuscript needs to be checked and rectified for grammatical flaws and logical flows.
4) The overview, basis, and description for the Tpr are missing in the abstract and introduction. Adding a clear notion and highlighting the previous reports on subcellular localization and roles of Tpr in normal and diseased brains will provide a clear lead for the novelty of the manuscript.
5) With reference to the interlink among nuclear pore complex (NPC) and Tpr in neural stem cells/progenitors in normal and AD brains, the rationality, existing lacuna/problems, and hypothesis have not clearly been established at the initial stage of the manuscript. Content from line number 47 to 73 in the introduction part demands good clarity for a better understanding.
6) The sample numbers are missing in all experiments in the methodology portion, though it has been mentioned in the figure legends. N=3 or N=4, is it sufficient?
7) In the methodology section, the information regarding the human brain samples from healthy and AD individuals is not satisfactory. The detail on the comorbid condition, treatment received, post-mortem detail, and the Braak system can also be explained in a bit clearer way. Also, The inclusion and exclusion criteria are missing.
8) In the aminal experimentation, the authors mentioned that only male animals were used, but in the human study why did you choose male and female?
9) In the abstract and introduction parts, indications for the use of post-human brain samples may provide additional value to the manuscript.
10) The ethical clearance for the animal study has not been mentioned in the methodology, but it has been declared in the additional info of the manuscript. However, institutional approval for experiments with the human brain is missing.
11) Why the brain samples were sectioned into two different μm (30- or 14-μm)? some statements are clueless or partial.
12) In the antigen retrieval step in human brain tissue, how was temperature 150°C achieved?
13) In the in-vitro study, authors isolated neural stem cells from SVZ of the mouse brain, while the entire manuscript is focused on the hippocampus. Authors need to justify or include the data from hippocampus-derived NSCs isolated as there are considerable differences exist between hippocampal NSC and NCSs from SVZ.
14) While the NSPCs were isolated from adult mice, the same appears to be missing from the AD animals for a better comparison of the disease-mediated expression of Trp and FG Nup in in-vitro.
15) The results sections need to be improved. As in the methodology section, the subtopics can be made separately for a better understanding.
16) The discussion needs major improvement. The interpretation of each result and supportive evidence, and scientific arguments for the therapeutic implementation of the results are missing.
17) The roles of Tpr and the nuclear pore complex jointly or independently of each other on the fate of NSC and the functional neurogenic program in health and disease/AD have not been clearly discussed.
Comments on the Quality of English Language1) The long and complex sentences need to be simplified in the overall text.
2) The manuscript needs to be checked and rectified for some grammatical flaws
Round 2
Reviewer 2 Report
Comments and Suggestions for Authors
Though the manuscript has been revised, the authors failed to supply and fulfill key basic information at the first instance. Many questions have not been addressed properly and overall major revision is not satisfactory. Considering the sample size from each experiment, the statistical assessments appear to be subtle. Data from the experiments with scarce human tissue samples and an unknown mix of females and males is not appropriate for publication. The authors declined to improve the overall quality and scientific soundness of the manuscript. The present version of the manuscript fails to meet the high standard of publication.
1) The ethical statement for using human samples is not clear. In addition, the authors did not reveal the full reference number (authorization) with the date/month/year of the approval for both human tissues and animal experiments which is not acceptable.
2) Concern regarding the number of samples used per experiment (animals, human samples, and in vitro experiments, number of single cells), the response made by authors is improper and not satisfactory. Tthe number of samples, used in all experiments, appears to be very low.
3) The number of samples used for human study (n = 2 healthy subjects and n=3 AD) appears to be improper which may not be considered for any statistical test. They did not reveal the sex of healthy and AD samples.
4) With reference to the sex difference, the answer from the authors is not reasonable.
“While male mice were used because female mice have a faster and earlier onset of Aβ plaque formation, the pathology of female and male mice is different” This will be implicated for humans too. Why did they choose both males and females in the human study? This will eventually add differences in the outcome of the results as the sample number is inadequate with a mix-up of both sexes. Thus, this data may not be valid.
5) All in vivo and in vitro data from animals, cells, and human samples need to be certified by an authorized professional statistician.
6) With reference to the thickness of the brain sections, the authors now mentioned in the method that Wild-type (WT) sections (30 μm) and AD sections (14μm) were further processed for free-floating and slide-mounted immunohistochemistry, respectively. In this case, control and AD tissues appear to be dissimilarly processed which may affect the outcome of the result. This may not be correct and not acceptable for the comparison of results.
7) Concerning the previous question no. 13 “In the in-vitro study, authors isolated neural stem cells from SVZ of the mouse brain, while the entire manuscript is focused on the hippocampus. The authors failed to address this issue. The justifications given by the authors are inappropriate.
8) The molecular size marker is missing in the immunoblots and the quality of immunoblots appears dull in the supplement.
9) Methodology, assessment, interpretation of results, and the establishment of hypothesis for the role of Tpr in the regulation of the neurogenic process in health and AD are not satisfactory.
10) The overall expression of language and clarity of some statements are not improved. The manuscript requires considerable effort for language editing.
Comments on the Quality of English LanguageStatements need major improvement